# Infinite-horizon Off-Policy Policy Evaluation with Multiple Behavior Policies

**Xinyun Chen**[1][†]**, Lu Wang**[2][†]**, Yizhe Hang**[3]**, Heng Ge**[4] **& Hongyuan Zha**[5][*]

[1] Insitute for Data and Decision Analytics, The Chinese University of Hong Kong, Shenzhen & Shenzhen Institute of Artificial Intelligence and Robotics for Society
[2] Department of Computer Science, East China Normal University
[3] Department of Computer Science, University of Science and Technology of China
[4] School of Mathematics and Statistics, Shandong University
[5] Insitute for Data and Decision Analytics, The Chinese University of Hong Kong, Shenzhen & Shenzhen Institute of Artificial Intelligence and Robotics for Society & Georgia Institute of Technology
[1]`chenxinyun@cuhk.edu.cn`, [2]`luwang@stu.ecnu.edu.cn`,
[3]`hangyhan@mail.ustc.edu.cn`, [4]`hengge@mail.sdu.edu.cn`,
[5]`zhahy@cuhk.edu.cn`

## Abstract

We consider off-policy policy evaluation when the trajectory data are generated by multiple behavior policies. Recent work has shown the key role played by the state or state-action stationary distribution corrections in the infinite horizon context for off-policy policy evaluation. We propose estimated mixture policy (EMP), a partially policy-agnostic methods to accurately estimate those quantities. With careful analysis, we show that EMP gives rise to estimates with reduced variance for estimating the state stationary distribution correction while it also offers a useful induction bias for estimating the state-action stationary distribution correction. In extensive experiments with both continuous and discrete environments, we demonstrate that our algorithm offers significantly improved accuracy compared to the state-of-the-art methods.

## 1 Introduction

In many real-world decision-making scenarios, evaluating a novel policy by directly executing it in the environment is generally costly and can even be downright risky. Examples include evaluating a recommendation policy (Swaminathan et al., 2017; Zheng et al., 2018), a treatment policy (Hirano et al., 2003; Murphy et al., 2001), and a traffic light control policy (Van der Pol & Oliehoek, 2016). Off-policy policy evaluation methods (OPPE) utilize a set of previously-collected trajectories (for example, website interaction logs, patient trajectories, or robot trajectories) to estimate the value of a novel decision-making policy without interacting with the environment (Precup et al., 2001; Dudík et al., 2011). For many reinforcement learning applications, the value of the decision is defined in a long- or infinite-horizon, which makes OPPE more challenging.

The state-of-the-art methods for infinite-horizon off-policy policy evaluation rely on learning *(discounted) state stationary distribution corrections* or *ratios*. In particular, for each state in the environments, these methods estimate the likelihood ratio of the long-term probability measure for the state to be visited in a trajectory generated by the *target policy*, normalized by the probability measure generated by the *behavior policy*. This approach can effectively avoid the exponentially high variance compared to the more classic importance sampling (IS) estimation methods (pre; Dudík et al., 2011; Hirano et al., 2003; Wang et al., 2017; Murphy et al., 2001), especially for infinite-horizon policy evaluation (Liu et al., 2018; Nachum et al., 2019; Hallak & Mannor, 2017). However, learning state stationary distribution requires detailed information on distributions of the behavior policy, and we call them *policy-aware* methods. As a consequence, policy-aware methods are difficult to apply

---

[*]Corresponding author. On leave from College of Computing, Georgia Institute of Technology.
[2]Equal Contribution.

when off-policy data are pre-generated by multiple behavior policies or when the behavior policy's form is unknown. To address this issue, Nachum et al. (2019) proposes a *policy-agnostic* method, DualDice, which learns the joint state-action stationary distribution correction that is much higher dimension, and therefore needs more model parameters than the state stationary distribution. Besides, there is no theoretic comparison between policy-aware and policy-agnostic methods.

In this paper, we propose a OPPE method with behavior policy learning, EMP (estimated mixture policy) for infinite-horizon off-policy policy evaluation with multiple known or unknown behavior policies. We call EMP a *partially policy-agnostic* method in the sense that, EMP does not require any information on each "physical" behavior policy, instead, it utilizes some aggregated information of the behavior policies learned from data. In detail, EMP includes a pre-estimation step using certain parametric model to learn a "virtual" policy (we call it the mixture policy and formally define it in Section 4). Hence, its performance depends on the accuracy of mixture policy estimation. Like the method in Liu et al. (2018), EMP obtain OPPE also via learning the state stationary distribution correction, so it remains computationally cheap and is scalable in terms of the number of behavior policies. Besides, inspired by Hanna et al. (2019), we provide a theoretic guarantee that EMP yields smaller mean square error (MSE) than the policy-aware methods in stationary distribution corrections learning, even in the single-behavior policy setting. On the other hand, compared to DualDice, EMP learns the state stationary distribution correction of smaller dimension, more importantly the estimation of the mixture policy can be considered as an inductive bias as far as the stationary distribution correction is concerned, and hence could achieve better performance when the pre-estimation is not expensive. In addition, we propose an ad-hoc improvement of EMP, whose theoretical analysis is left for future studies. EMP is compared with both policy-aware and policy-agnostic methods in a set of continuous and discrete control tasks and shows significant improvement.

## 2 BACKGROUND AND RELATED WORK

We first introduce the general setting of OPPE in infinite horizon. Then we review two families of OPPE methods, based on importance sampling (IS) and stationary distribution correction learning, respectively.

### 2.1 INFINITE-HORIZON OFF-POLICY POLICY EVALUATION

We consider a Markov Decision Process (MDP) and our goal is to estimate the infinite-horizon *average reward*. The environment is specified by a tuple $\mathcal{M} = \langle S, A, R, T \rangle$, consisting of a state space, an action space, a reward function, and a transition probability function. A policy $\pi$ interacts with the environment iteratively, starting with an initial state $s_0$. At step $n = 0, 1, \dots$ , the policy produces a distribution $\pi(\cdot|s_n)$ over the actions $A$, from which an action $a_n$ is sampled and applied to the environment. The environment stochastically produces a scalar reward $r(s_n, a_n)$ and a next state $s_{n+1} \sim T(\cdot|s_n, a_n)$. The infinite-horizon average reward under policy $\pi$ is

$$R_\pi = \lim_{N \to \infty} \frac{1}{N+1} \sum_{n=0}^{N} \mathbb{E}_\pi \left[ r(s_n, a_n) \right].$$

Without gathering new data, off-policy policy evaluation (OPPE) considers the problem of estimating the expected reward of a target policy $\pi$ via pre-collected state-action-reward tuples from policies that are different from $\pi$, which are called behavior policies. In our paper, we consider the general setting that the data are generated by multiple behavior policies $\pi_j (j = 1, .., m)$. Most OPPE literature has focused on the single-behavior-policy case where $m = 1$. In this case, we denote the behavior policy by $\pi_0$ to distinguish from the multiple-behavior-policy case. Roughly speaking, most OPPE methods can be grouped into two categories: importance-sampling(IS) based OPPE and stationary-distribution-correction based OPPE.

### 2.2 IMPORTANCE SAMPLING POLICY EVALUATION

As for short-horizon off-policy policy evaluation, importance sampling policy evaluation (IS) methods (Precup et al., 2001; Dudík et al., 2011; Swaminathan et al., 2017; Precup et al., 2000; Horvitz &

Thompson, 1952) have shown promising empirical results. The main idea of importance sampling based OPPE is using importance weighting $\pi/\pi_j$ to correct the mismatch between the target policy $\pi$ and the behavior policy $\pi_j$ that generates the trajectory.

One key element in our EMP method are inspired by importance sampling literature. Li et al. (2015) and Hanna et al. (2019) show that using estimated behavior policy in the importance weighting can reduce the mean square error (MSE). EMP also uses estimated policy, but there are two key difference between EMP and the previous works: (1) EMP is not an IS-based method, it involves a min-max problem; (2) EMP focuses on multiple-behavior-policy setting while these papers have focused on single-behavior setting.

## 2.3 Policy Evaluation via Learning Stationary Distribution Correction

The state-of-the-art methods for long-horizon off-policy policy evaluation are stationary-distribution-correction based (Liu et al., 2018; Nachum et al., 2019; Hallak & Mannor, 2017). Let $d_{\pi_0}(s)$ and $d_\pi(s)$ be the stationary distribution of state $s$ under the behavior policy $\pi_0$ and target policy $\pi$ respectively. The main idea of such methods is directly applying importance weighting by $\omega = d_\pi/d_{\pi_0}$ on the stationary state-visitation distributions to avoid the exploding variance suffered from IS, and estimate the average reward as

$$R_\pi = \mathbb{E}_{(s,a)\sim d_\pi}[r(s,a)] = \mathbb{E}_{(s,a)\sim d_{\pi_0}}\left[\omega(s)\cdot\frac{\pi(a|s)}{\pi_0(a|s)}r(s,a)\right]. \tag{1}$$

For example, Liu et al. (2018) uses min-max approach to estimate $\omega$ directly from the data. This class of methods require exact knowledge of behavior policy $\pi_0$ and are not straightforward to apply in multiple-behavior-policy setting. Recently, Nachum et al. (2019) proposed DualDice to overcome such limitation by learning the state-action stationary distribution correction $\omega(s,a) = d_\pi(s)\pi(a|s)/d_{\pi_0}(s)\pi_0(a|s)$.

## 3 Single Behavior Policy Estimation

When the behavior policy $\pi_0$ in (1) is unknown, a natural idea is to estimate it from data. In this section, we focus on the standard case where the data are generated by a single behavior policy so that estimation of behavior policy is more straightforward. We first breifly review the method introduced by Liu et al. (2018) in Section 3.1, which we shall refer as the BCH method in the rest of the paper, to explain the min-max problem formulation of the stationary distribution correction learning task. In Section 3.2, we show that behavior policy estimation is beneficial in two aspects. First, it extends the stationary distribution correction method to settings where behavior policy is unknown. Second, even when the behavior policy is known with exact values, we prove that the stationary distribution correction learned using behavior policy estimation has smaller MSE than that using exact values. Later, we will extend this behavior policy estimation idea to more general multiple-behavior-policy cases in Section 4.

### 3.1 Learning Stationary Distribution Correction with Exact Behavior Policy

Assume the data, consisting of state-action-next-state tuples, are generated by a single behavior policy $\pi_0$, i.e. $\mathcal{D} = \{(s_n, a_n, s'_n) : n = 1, 2, ..., N\}$. Recall that $d_{\pi_0}$ and $d_\pi$ are the stationary state distribution under the behavior and target policy respectively, and $\omega = d_\pi/d_{\pi_0}$ is the *stationary distribution correction*. In the rest of Section 3, by slight notation abuse, we also denote $d_\pi(s,a) = d_\pi(s)\pi(a|s)$, $d_{\pi_0}(s,a) = d_{\pi_0}(s)\pi_0(a|s)$ and $d_{\pi_0}(s,a,s') = d_{\pi_0}(s)\pi_0(a|s)T(s'|a,s)$.

We briefly review the BCH method proposed by Liu et al. (2018). As $d_\pi(s)$ is the stationary distribution of $s_n$ as $n \to \infty$ under policy $\pi$, it follows that:

$$d_\pi(s') = \sum_{s,a} d_\pi(s)\pi(a|s)T(s'|s,a) = \sum_{s,a} \omega(s)\frac{\pi(a|s)}{\pi_0(a|s)}d_{\pi_0}(s)\pi_0(a|s)T(s'|a,s), \quad \forall s'.$$

Therefore, for any function $f : S \to \mathbb{R}$,

$$\sum_{s'} \omega(s')d_{\pi_0}(s')f(s') = \sum_{s,a,s'} \omega(s)\frac{\pi(a|s)}{\pi_0(a|s)}d_{\pi_0}(s)\pi_0(a|s)T(s'|a,s)f(s').$$

Recall that $d_{\pi_0}(s, a, s') = d_{\pi_0}(s)\pi_0(a|s)T(s'|a, s)$, so $\omega$ and the data sample satisfy the following equation

$$\mathbb{E}_{(s,a,s') \sim d_{\pi_0}} \left[ \left( \omega(s') - \omega(s) \frac{\pi(a|s)}{\pi_0(a|s)} \right) f(s') \right] = 0, \forall f.$$

BCH solves the above equation via the following min-max problem:

$$\min_\omega \max_f \; \mathbb{E}_{(s,a,s') \sim d_{\pi_0}} \left[ \left( \omega(s') - \omega(s) \frac{\pi(a|s)}{\pi_0(a|s)} \right) f(s') \right]^2, \tag{2}$$

and use *kernel method* to solve $\omega$. The derivation of kernel method is put in Appendix A.

## 3.2 LEARNING STATIONARY DISTRIBUTION CORRECTION WITH ESTIMATED BEHAVIOR POLICY

The objective function in the min-max problem (2), evaluated by data sample, can be viewed as a one-step importance sampling estimation. As shown in Hanna et al. (2019), importance sampling with estimated behavior policy has smaller MSE. Motivated by this fact and the heuristic that better objective function evaluation will lead to more accurate solution, we show that the BCH method can also be improved by using estimated behavior policy to obtain smaller asymptotic MSE. We will use this result to build theoretic guarantee for the performance of EMP method in Section 4.

To formally state the theoretic result, we need to introduce more notation. Assume that we are given a class of stationary distribution correction $\Omega = \{\omega(\eta; s) : \eta \in \mathcal{E}_\eta\}$, and there exists $\eta_0 \in \mathcal{E}_\eta$ such that the true distribution correction $\omega(s) = \omega(\eta_0; s)$. Let $\omega(\tilde{\eta}; s)$ be the stationary distribution correction learned by the min-max problem (2) and $\omega(\hat{\eta}; s)$ be that learned by a min-max problem similar to (2) with $\pi_0$ replaced with its estimation $\hat{\pi}_0$. Intuitively, $\hat{\pi}_0$ is estimated from the data sample and appears in the denominator, as a result, it could cancel out a certain amount of random error in data sample. Following this intuition and applying the proof techniques in Henmi et al. (2007), we establish the following theoretic guarantee that using estimated behavior policy [1] yields better estimates of the stationary distribution correction.

**Theorem 1.** *Under Assumptions 1 and 2, we have, asymptotically*

$$E[(\hat{\eta} - \eta_0)^2] \leq E[(\tilde{\eta} - \eta_0)^2].$$

As a direct consequence, we derive the finite-sample error bound for $\hat{\eta}$.

**Corollary 1.** *Let $N$ be the number of $(s, a, s')$ tuples in the data. Under Assumptions 1 and 2,*

$$\mathbb{E}[(\hat{\eta} - \eta_0)^2] = O\left(\frac{1}{N}\right)$$

.

Due to the space limit, we put the precise descriptions of Assumptions 1 and 2 (which involves details of the kernel method that solves the min-max problem (2)) for Theorem 1 and Corollary 1 to hold and their proofs are in Appendix B.

## 4 EMP FOR MULTIPLE BEHAVIOR POLICIES

In this section, we apply the behavior policy estimation idea and develop our EMP method for OPPE in settings of multiple behavior policies and establish theoretic variance reduction results for EMP. In Section 4.1, we first clarify what is the policy to estimate from data when there are multiple behavior policies. Then, we introduce EMP in Section 4.2 and establish variance reduction result in 4.3.

## 4.1 MIXTURE POLICY AND MIXTURE STATE DISTRIBUTION

Let's first take a closer look at the distribution of data generated by multiple behavior policies. In particular, we show that the data from different behavior policies can be pooled together as if they are generated by a virtual"mixture policy" $\pi_M$, which plays a key role in derivation of EMP method.

---

[1] We use MLE to obtain $\hat{\pi}_0$.

In the multiple-behavior setting, we assume the state-action-next-state tuples are generated by $m$ different unknown behavior policies $\pi_j, j = 1, 2, ..., m$. For each $j$, there are $N_j$ state-action-next-state tuples generated by $\pi_j$ and follows the corresponding stationary state distribution $d_{\pi_j}$. Let $N = \sum_j N_j$ and denote by $w_j = N_j/N$ the proportion of data generated by policy $\pi_j$. We use $\mathcal{D}_M$ to denote the data set, then $\mathcal{D}_M = \{(s_{j,n_j}, a_{j,n_j}, s'_{j,n_j}) : j = 1, 2, .., m, n_j = 1, 2, ..., N_j\}$. Note that the policy label $j$ in the subscript is only for notation clarity and it is not revealed in the data. Then, if we randomly draw a single $(s, a, s')$ tuple from $\mathcal{D}_M$, its distribution function is the mixture of state-action-next-state tuple distributions generated by each behavior policy:

$$d_M(s, a, s') := \sum_j w_j d_{\pi_j}(s) \pi_j(a|s) T(s'|a, s).$$

With slight notation abuse, we write $d_M(s) = \sum_j w_j d_{\pi_j}$ as the **mixture state distribution**. For each state-action pair $(a, s)$, define the **mixture policy** $\pi_M(a|s)$ as the weighted average of the behavior policies:

$$\pi_M(a|s) := \sum_j \frac{w_j d_{\pi_j}(s)}{d_M(s)} \pi_j(a|s), \forall (s, a). \tag{3}$$

The following result shows that, the multiple-behavior-policy data $\mathcal{D}_M$ and the corresponding state distribution $d_M(s)$ can be viewed as if they were generated by the mixture policy $\pi_M$

**Proposition 1.** *The state-action-next-station tuples generated by $\pi_M$ follows the distribution $d_M(s, a, s')$. As a consequence, $d_M(s)$ is the stationary state distribution generated by $\pi_M$.*

Let $\omega_M(s) = d_\pi(s)/d_M(s)$ be the likelihood ratio of the state distribution generated by the target policy over the mixture state distribution. We can estimate the average reward by $\omega_M$:

**Proposition 2.** *The average award satisfies*

$$R_\pi = E_{(s,a) \sim d_M} \left[ \omega_M(s) \frac{\pi(a|s)}{\pi_M(a|s)} r(s, a) \right]. \tag{4}$$

## 4.2 EMP METHOD

Proposition 1 shows that $d_M(s)$, the state distribution generated by multiple behavior policies, is equal to the state stationary distribution generated by $\pi_M$. If $\pi_M$ is known, then, following the same argument of BCH, we can learn $\omega_M(s) = \omega(\eta_0; s)$ via a similar min-max problem as (2), by replacing $d_{\pi_0}$ with $d_M$ and $\pi_0$ with $\pi_M$, and then, estimate $R_\pi$ using (4), by replacing $d_{\pi_0}$ with $d_M$, $\pi_0$ with $\pi_M$ and $\omega$ with $\omega_M$. But $\pi_M$ is usually unknown. Indeed, it involves not only the behavior polices $\pi_j$, but also their stationary distributions $d_{\pi_j}$, which are unknown and hard to compute. In our EMP method, we will estimate $\pi_M$ directly from data, without learning $\pi_j$ and $d_{\pi_j}$. In particular, we assume the mixture policy $\pi_M$ belongs to some parametric family, i.e. $\pi_M(a|s) = \pi(\theta_M; a, s)$. For instance, $\pi(\theta; a, s)$ could be a regression model or a neural network. We then estimate $\theta_M$ by MLE, i.e.

$$\hat{\theta}_M = \arg \max_\theta \sum_j \sum_{n=1}^{N_j} \log(\pi(\theta; s_{j,n}, a_{j,n})). \tag{5}$$

After this pre-estimation step, we replace the exact mixture policy $\pi_M$ with the estimated mixture policy $\pi(\hat{\theta}_M; \cdot)$ and finally formulate the following min-max problem for EMP to learn $\omega_M$:

$$\min_\eta \max_f \mathbb{E}_{(s,a,s') \sim \mathcal{D}_M} \left[ \left( \omega(\eta; s') - \omega(\eta; s) \frac{\pi(a|s)}{\pi(\hat{\theta}_M; a, s)} \right) f(s') \right]. \tag{6}$$

Applying Theorem 1, we build the following MSE bound for EMP method.

**Theorem 2.** *Under the same conditions of Theorem 1, if $\omega(\tilde{\eta}; s)$ and $\omega(\hat{\eta}; s)$ are the stationary distribution correction learned from (6) and from the same min-max problem but with exact value of $\pi_M$, then, asymptotically*

$$E[(\hat{\eta} - \eta_0)^2] \leq E[(\tilde{\eta} - \eta_0)^2].$$

*As a result, $E[(\hat{\eta} - \eta_0)^2] = O\left(\frac{1}{N}\right)$.*

### 4.3 WHY POOLING IS BENEFICIAL FOR EMP

One important feature of EMP is that it pools the data from different policy behaviors together and treat them as if they are from a single mixture policy. Of course, pooling makes EMP applicable to settings with minimal information on the behavior policies, for instance, EMP does not even require the knowledge on the number of behavior policies. In this part, we show that, the pooling feature of EMP is not just a compromise to the lack of behavior policy information, it also leads to variance reduction in an intrinsic manner.

If instead, the data are treated separately according to the behavior policies, we can still use EMP, or any OPPE method for single behavior policy, to obtain the stationary distribution correction $\omega_j = d_\pi / d_{\pi_j}$ for each behavior policy. Given $\omega_j$, a common approach for variance reduction is to apply multiple importance sampling (MIS) (Tirinzoni et al., 2019; Veach & Guibas, 1995) technique and the average reward estimator is of the form

$$\hat{R}_{MIS} = \sum_{j=1}^{m} \frac{1}{N_j} \sum_{n=1}^{N_j} h_j(s_{j,n}) \omega_j(s_{j,n}) r_\pi(s_{j,n}), \text{ with } r_\pi(s) = \sum_a \pi(a|s) r(s,a). \quad (7)$$

where the function $h$ is often referred to as heuristics and must be a partition of unity, i.e., $\sum_j h_j(s) = 1$ for all $s \in S$. It has been proved by Veach & Guibas (1995) that MIS is unbiased, and, for given $w_j = N_j / N$, there is an optimal heuristic function to minimize the variance of $\hat{R}_{MIS}$.

**Proposition 3.** *(Veach & Guibas, 1995) For MIS with fixed values of $w_j$, $j = 1, 2, ..., m$, among all possible values of heuristics $h$, the balanced heuristic*

$$h_j(s) = \frac{w_j d_{\pi_j}(s)}{\sum_{j=1}^{m} w_j d_{\pi_j}(s)}, \ \forall j = 1, 2, ..., m \ and \ s \in S,$$

*reaches the minimal variance.*

Plug the optimal heuristic $h_j(s)$ into MIS estimator (7), and we will obtain that the optimal MIS estimator coincides with the EMP estimator (4). In this light, by pooling the data together and directly learning $\omega_M$, EMP also learns the optimal MIS weight inexplicitly.

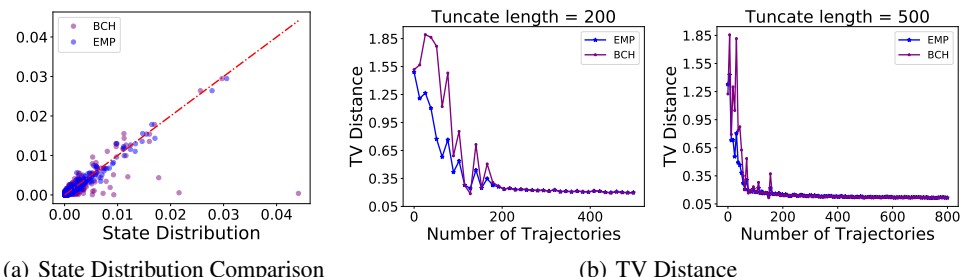

(a) State Distribution Comparison  (b) TV Distance

Figure 1: (a) shows that scatter plot of pairs $(\hat{d}_{\pi_{\text{true}}}, d_\pi)$ and pairs $(\hat{d}_{\pi_{\text{esti}}}, d_\pi)$. The diagonal line indicates exact estimation. The default values of the number of trajectories is 200, and the length of horizon is 200. (b) shows the weighted total variation distance (TV distance) between $\hat{d}_{\pi_{\text{true}}}$ and $d_\pi$, $\hat{d}_{\pi_{\text{esti}}}$ and $d_\pi$ respectively, along different number of trajectories and the length of horizons.

## 5 EXPERIMENT

In this section, we evaluate EMP on OPPE problems in three discrete-control tasks Taxi, Singlepath, Gridworld and one continuous-control task Pendulum (see Appendix D.1 for the details), in both single-behavior-policy (Section 5.1)and multiple-behavior-policy settings (Section 5.2), with following purposes: (i) to compare the performance of EMP with existing OPPE methods; (ii) to validate the theoretical properties for EMP; (iii) to explore potential improvement of EMP methods for future study. We will release the codes with the publication of this paper for relevant study.

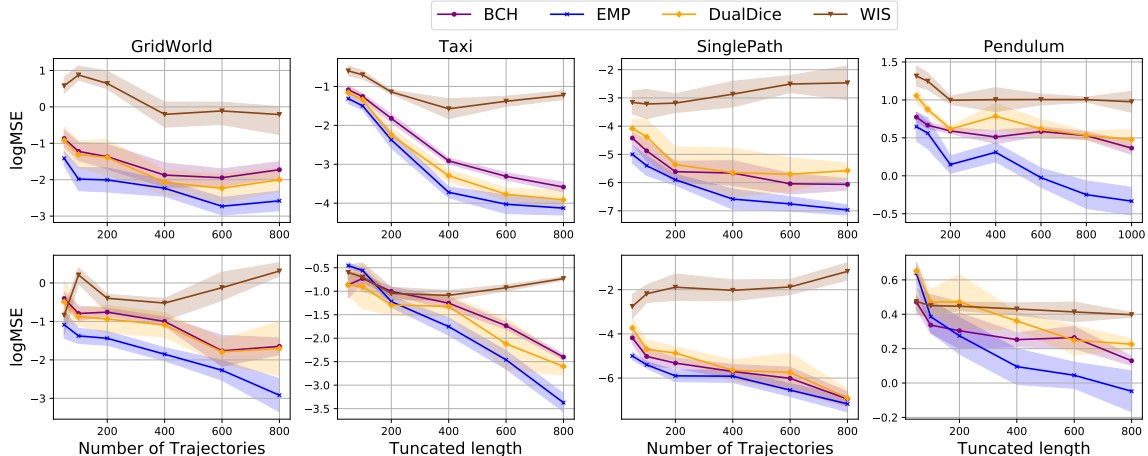

Figure 2: Single-behavior-policy results of BCH, EMP, DualDice and WIS across continuous and discrete environments with average reward. Each node indicates the mean value and the bars represents the standard error of the mean.

## 5.1 RESULTS FOR SINGLE BEHAVIOR POLICY

In this section, we compare the EMP method with BCH, DualDice[2] and step-wise importance sampling (IS) in the setting of single-behavior policy, i.e. the data is generated from a single behavior policy.

**Experiment Set-up.** A single behavior policy is learned by a certain reinforcement learning algorithm [3] for evaluating BCH and IS. This single behavior policy then generates a set of trajectories consisting of s-a-s-r tuples. These tuples are used to estimate the behaviour policy in EMP method [4] as well as estimating the stationary distribution corrections for estimating the average step reward of the target policy.

**Stationary Distribution Learning Performance.** To validate Theorem 1, we use the Taxi domain as an example to compare the stationary distribution $\hat{d}_{\pi_{\text{true}}}$ and $\hat{d}_{\pi_{\text{esti}}}$ learned by BCH (using exact behavior policy) and EMP (using estimated behavior policy). Figure 1(a) plots the scatter pairs $(\hat{d}_{\pi_{\text{true}}}, d_{\pi})$ and $(\hat{d}_{\pi_{\text{esti}}}, d_{\pi})$ estimated by 200 trajectories of 200 steps. It shows that $\hat{d}_{\pi_{\text{esti}}}$ approximate $d_{\pi}$ better than $\hat{d}_{\pi_{\text{true}}}$. Figure 1(b) and Figure 1(b) compare the TV distance from $\hat{d}_{\pi_{\text{true}}}$ and $\hat{d}_{\pi_{\text{esti}}}$ to $d_{\pi}$. The results indicate that both $\hat{d}_{\pi_{\text{true}}}$ and $\hat{d}_{\pi_{\text{esti}}}$ converge, while $\hat{d}_{\pi_{\text{esti}}}$ converges faster and is significantly closer to $d_{\pi}$ when the data size is small. These observations are well consistent with Theorem 1.

**Policy Evaluation Performance.** Figure 2 reports the MSE of policy evaluation by EMP, BCH, DualDice and IS methods for the 4 different environments. We observe that, (i) EMP consistently obtains smaller MSE than the other three methods for different sample scales and different environments. (ii) The performance of EMP, BCH and DualDice improves as the number of trajectories and length of horizons increase, while the IS method suffers from growing variance.

## 5.2 RESULTS FOR MULTIPLE UNKNOWN BEHAVIOR POLICIES

In this section, we compare the performance of EMP with a multiple-behavior version of BCH method, DualDice and MIS (Precup et al., 2000), and explore potential improvement of EMP methods.

---

[2]DualDice was actually designed for discounted problems, not for the average problem as considered in this paper. However, it is the only policy-agnostic algorithm for off-policy evaluation in literature, to the best of our knowledge.

[3]We use Q-learning in discrete control tasks and Actor Critic in continuous control tasks.

[4]For discrete state-action space, we estimate the behavior policy by count-frequency. For continuous state-action space, we fit the behavior policy by a neural network model and use MLE to estimate the model parameters. The details are given in Appendix D.2.

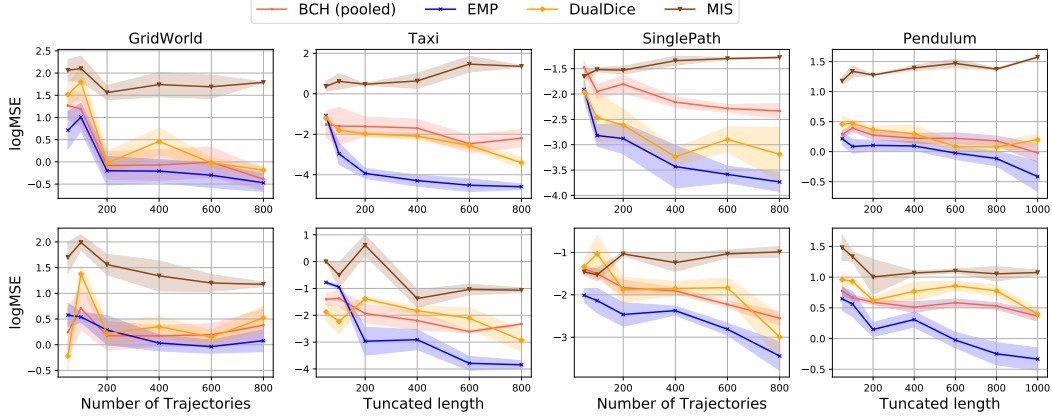

Figure 3: Multiple-behavior-policy results of BCH (pooled), EMP, DualDice and MIS across continuous and discrete environments with average reward.

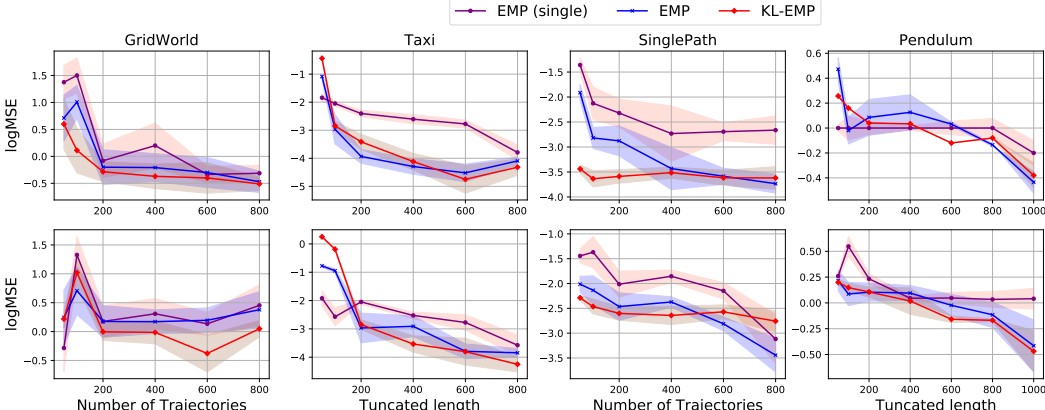

Figure 4: Multiple-behavior-policy results of EMP (single), EMP, KL-EMP across continuous and discrete environments with average reward.

**Experiment Setup.** For each environment, we use 5 behavior policies to generate the data. They are obtained by reinforcement learning after different number of training steps, i.e. the first 20%, 40%, 50%, 70%, 90% of training steps by which we learn the target policy. To form the multiple-behavior-policy samples, we generate same number of trajectories of equal length from each policy.

**Policy Evaluation Performance.** We implement 4 methods: EMP, BCH, DualDice and WIS introduced in pre using balanced heuristics. To compare with policy-aware BCH in the multiple-behavior-policy setting, we develop an extension BCH method called BCH (pooled), which utilize the exact values of all the behavior policies[5]. Figure 3 shows that, in all 4 environments, we found EMP consistently obtain smaller MSE than the other three methods.

**Pooling is Beneficial.** To validate the variance reduction result Proposition 3, we compare EMP with its variation EMP (single) in which trajectories from different behaviors are not pooled. In detail, EMP (single) applies EMP to each group of data generated by same behavior policy and return the mean average reward estimation. Figure 4 shows that EMP outperforms EMP (single), which is consistent with the theoretic variance reduction result. On the other hand, the optimality of EMP in Proposition 3 holds for fixed $w_j$, we now explore the possibility of further variance reduction via optimizing $w_j$, i.e. , by re-weighting the proportion of samples generated by different behavior policies. In detail, we implement a variation of EMP in which the data samples are re-weighted according to the KL-divergence between its behavior policy and the target policy[6]. Figure 4 also shows that the performance of KL-EMP has greater improvement with the increase of sample size and could outperform EMP in cases of large sample size where the KL-divergence is better estimated.

## 6 CONCLUSION

In this paper, we advocate the viewpoint of partial policy-awareness and the benefits of estimating a "virtual" mixture policy for off-policy policy evaluation. The theoretical results of reduced variance coupled with experimental results illustrate the power of this class of methods. One key question that still remains is the following: if we are willing to estimate the individual behavior policies, can we further improve EMP by developing an efficient algorithm to compute the optimal weights? The preliminary experiment results suggest that the answer would be yes, and we will leave this for future study.

**Acknowledgement:** We thank Zhaoyuan Li for her helpful technical comments on the proofs. Xinyun Chen is grateful to the financial support from NSFC Grant No. 91646206 and 11901493. Part of the work done by Hongyuan Zha is supported by Shenzhen Institute of Artificial Intelligence and Robotics for Society, and Shenzhen Research Institute of Big Data.

ACKNOWLEDGMENTS

We thank Zhaoyuan Li for her helpful technical comments on the proofs. Xinyun Chen is grateful to the financial support from NSFC Grant No. 91646206 and 11901493. Part of the work done by Hongyuan Zha is supported by Shenzhen Institute of Artificial Intelligence and Robotics for Society, and Shenzhen Research Institute of Big Data.

---

[5]The details of BCH (pooled) are given in Appendix D.4
[6]The details of KL-EMP are given in Appendix D.3

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

## A  KERNEL METHOD

We use the reproducing kernel Hilbert space to solve the mini-max problem of BCH (Liu et al. (2018)). The key property of RKHS we leveraged is called **reproducing property**. The reproducing property claims, for any function $f \in \mathcal{H}$ ($\mathcal{H}$ is a RKHS), the evaluation of $f$ at point x equals its inner product with another function in RKHS: $f(s) = \langle f, k(s, \cdot) \rangle_{\mathcal{H}}$.

Given the objective function of BCH $L(w, f) = \mathbb{E}_{(s,a,s') \sim d_{\pi_0}}[(\omega(s)\frac{\pi(a|s)}{\pi_0(a|s) - \omega})f(s')]$. We use the reproducing property to obtain the closed form representation of $\max_{f \in \mathcal{F}} L(w, f)^2$, which is shown as follows:

$$\max_{f \in \mathcal{F}} L(w, f)^2 = \mathbb{E}_{(s,a,s') \sim d_{\pi_0}, (\bar{s},\bar{a},\bar{s}') \sim d_{\pi_0}} \left[ \Delta\left(\omega; s, a, s'\right) \Delta\left(w; \bar{s}, \bar{a}, \bar{s}'\right) k\left(s', \bar{s}'\right) \right]$$

.

This equation has been proved in BCH Liu et al. (2018).

## B  PROOF OF THEOREM 1

### B.1  ASSUMPTIONS

In this appendix, we provide the mathematical details and proof of Theorem 1. We first introduce some notations and assumptions.

We assume the behavior policy $\pi_0(a|s)$ belongs to a class of policies $\Pi = \{\pi(\theta; a, s) : \theta \in \mathcal{E}_\theta\}$, where $\mathcal{E}_\theta$ is the parameter space, i.e. there exists $\theta_0 \in \mathcal{E}_\theta$ such that $\pi_0(a|s) = \pi(\theta_0; a, s)$. The estimated behavior policy $\hat{\pi}_0(a|s) = \pi(\hat{\theta}; a, s)$ is obtained via maximum likelihood method, i.e.

$$\hat{\theta} = \arg\max \sum_{n=1}^{N} \log(\pi(\theta; s_n, a_n)).$$

We assume central limit theorem holds for $\hat{\theta}$:

**Assumption 1.**  *(CLT of MLE)*

$$\mathbb{E}_{\mathcal{D}}[(\hat{\theta} - \theta_0)^2] = O(1/N).$$

Recall that we have assumed in Section 3.2 that the true stationary distribution correction $\omega(s) = \omega(\eta_0; s)$. Using the kernel method introduced in Appendix A, we estimate $\hat{\omega}(s) = \omega(\hat{\eta}; s)$ by

$$\hat{\eta} = \arg\min_{\eta} \sum_{1 \le i,j \le N} G(\eta, \hat{\theta}; x_i, x_j),$$

with $x_i = (s_i, a_i, s_i')$ and

$$G(\eta, \hat{\theta}; (x_i, x_j)) = \left( \omega(\eta; s_i) \frac{\pi(a_i|s_i)}{\pi(\hat{\theta}, a_i, s_i)} - \omega(\eta, s_i') \right) \left( \omega(\eta; s_j) \frac{\pi(a_j|s_j)}{\pi(\hat{\theta}, a_j, s_j)} - \omega(\eta, s_j') \right) k(s_i', s_j').$$

The BCH method estimate $\tilde{\theta}$ by

$$\tilde{\eta} = \arg\min_{\eta} \sum_{1 \le i,j \le N} G(\eta, \theta_0; x_i, x_j).$$

**Assumption 2.**  *We assume the following regularity conditions on $G$:*

1. *$G$ is second order differentiable.*

2. *$\mathbb{E}[\partial_\eta \partial_\theta G(\eta_0, \theta_0; x_i, x_j)]$ is finite.*

3. *$\mathbb{E}[\partial_\eta^2 G(\eta_0, \theta_0; x_i, x_j)]$ is finite and non-zero.*

4. *$\mathbb{E}[\partial_\eta G(\eta_0, \theta_0; x_i, x_j)^2]$ is finite.*

Here we simply write $\mathbb{E}_{x_i \sim d_{\pi_0}, x_j \sim d_{\pi_0}}$ as $\mathbb{E}$ for the simplicity of notation.

## B.2 PROOF OF THEOREM 1

**Theorem 1.** *Under Assumptions 1 and 2, we have, asymptotically*
$$E[(\hat{\eta} - \eta_0)^2] \le E[(\tilde{\eta} - \eta_0)^2].$$

*Proof.* Following the kernel method,
$$\hat{\eta} = \arg\min_\eta = \arg\min_\eta \sum_{1 \le i,j \le N} G(\eta, \hat{\theta}; (x_i, x_j)) \text{ with } x_i = (s_i, a_i, s_i') \text{ and } s_i' \triangleq s_{i+1}.$$

Then, $\sum_{1 \le i,j \le N} \partial_\eta G(\hat{\eta}, \hat{\theta}; (x_i, x_j)) = 0$, we have

$$0 = \frac{1}{N\sqrt{N}} \sum_{1 \le i,j \le N} \partial_\eta G(\eta_0, \theta_0; (x_i, x_j)) + \sqrt{N}(\hat{\eta} - \eta_0)\frac{1}{N^2} \sum_{1 \le i,j \le N} \partial_\eta^2 G(\eta_0, \theta_0; (x_i, x_j))$$

$$+ \sqrt{N}(\hat{\theta} - \theta)\frac{1}{N^2} \sum_{1 \le i,j \le N} \partial_\theta \partial_\eta G(\eta_0, \theta_0; (x_i, x_j))$$

$$= \frac{1}{N\sqrt{N}} \sum_{1 \le i,j \le N} \partial_\eta G(\eta_0, \theta_0; (x_i, x_j)) + \sqrt{N}(\hat{\eta} - \eta_0)\mathbb{E}\left[\partial_\eta^2 G(\eta_0, \theta_0; (x_1, x_2))\right]$$

$$+ \sqrt{N}(\hat{\theta} - \theta)\mathbb{E}\left[\partial_\theta \partial_\eta G(\eta_0, \theta_0; (x_1, x_2))\right] + o_p(1).$$

By definition, $\tilde{\eta}$ is the solution to the optimization problem$\max_\eta \sum_{1 \le i,j \le N} G(\eta, \theta_0; (x_i, x_j))$. Therefore, $0 = \frac{1}{N\sqrt{N}} \sum_{1 \le i,j \le N} \partial_\eta G(\eta_0, \theta_0; (x_i, x_j)) + \sqrt{N}(\tilde{\eta} - \eta_0)\mathbb{E}\left[\partial_\eta^2 G(\eta_0, \theta_0; (x_1, x_2))\right] + o_p(1)$. Define $S(\theta; (x_i, x_j)) = \log(\pi(\theta; s_i, a_i)) + \log(\pi(\theta; s_j, a_j))$. Similarly, $\hat{\theta}$ is the optimal solution to $\arg\max_\theta \sum_{1 \le i,j \le N} S(\theta; (x_i, x_j))$. Therefore, $0 = \frac{1}{N\sqrt{N}} \sum_{1 \le i,j \le N} \partial_\theta S(\theta_0; (x_i, x_j)) + \sqrt{N}(\hat{\theta} - \theta_0)\mathbb{E}\left[\partial_\theta^2 S(\theta_0; (x_1, x_2))\right] + o_p(1)$. Following the proof of Theorem 1 of (Henmin et al. 2007), it suffices to prove that

$$\mathbb{E}\left[\partial_\theta \partial_\eta G(\eta_0, \theta_0; (x_1, x_2))\right] = \mathbb{E}\left[-\partial_\eta G(\eta_0, \theta_0; (x_1, x_2))\partial_\theta S(\theta_0; (x_1, x_2))\right]. \tag{8}$$

One can check
$$\mathbb{E}\left[\partial_\eta G(\eta_0, \theta_0; (x_1, x_2))\right]$$

$$= \mathbb{E}\left[k(s_1', s_2')\left[\left(\partial_\eta \omega(\eta_0; s_1)\frac{\pi(a_1|s_1)}{\pi(\theta_0; a_1, s_1)} - \partial_\eta \omega(\eta_0; s_1')\right)\left(\omega(\eta_0; s_2)\frac{\pi(a_2|s_2)}{\pi(\theta_0; a_2, s_2)} - \omega(\eta_0; s_2')\right)\right.\right.$$

$$\left.\left. + \left(\partial_\eta \omega(\eta; s_2)\frac{\pi(a_2|s_2)}{\pi(\theta_0; a_2, s_2)} - \partial_\eta \omega(\eta_0; s_2')\right)\left(\omega(\eta_0; s_1)\frac{\pi(a_1|s_1)}{\pi(\theta_0; a_1, s_1)} - \omega(\eta_0; s_1')\right)\right]\right]$$

$$= \mathbb{E}\left[(k(s_1', s_2') + k(s_2', s_1'))\left(\partial_\eta \omega(\eta_0; s_1)\frac{\pi(a_1|s_1)}{\pi(\theta_0; a_1, s_1)} - \partial_\eta \omega(\eta_0; s_1')\right)\left(\omega(\eta_0; s_2)\frac{\pi(a_2|s_2)}{\pi(\theta_0; a_2, s_2)} - \omega(\eta_0; s_2')\right)\right]$$

The last equality holds because $(x_1, x_2) \sim d_{\pi_0}(s_1)\pi(x_1; \eta_0) \otimes d_{\pi_0}(s_2)\pi(x_2; \eta_0)$. For the simplicity of derivation, in the rest of proof, we denote

$$g_1 = \partial_\eta \omega(\eta_0; s_1)\frac{\pi(a_1|s_1)}{\pi(\theta_0; a_1, s_1)} - \partial_\eta \omega(\eta_0; s_1'), \ g_2 = \omega(\eta_0; s_2)\frac{\pi(a_2|s_2)}{\pi(\theta_0; a_2, s_2)} - \omega(\eta_0; s_2').$$

On the other hand, note that

$$\partial_\theta S(\theta; (x_1, x_2)) = \frac{\partial_\theta \pi(\theta; a_1, s_1)}{\pi(\theta; a_1, s_1)^2} + \frac{\partial_\theta \pi(\theta; a_2, s_2)}{\pi(\theta; a_2, s_2)^2}.$$

Then, we derive
$$\mathbb{E}\left[\partial_\theta \partial_\eta G(\eta_0, \theta; (x_1, x_2))\right]$$

$$= \mathbb{E}\left[(k(s_1', s_2') + k(s_2', s_1'))\left[-\partial_\eta \omega(\eta_0; s_1)\frac{\partial_\theta \pi(\theta_0; a_1, s_1)}{\pi(\theta_0; a_1, s_1)^2} \cdot g_2 - \omega(\eta_0; s_2)\frac{\partial_\theta \pi(\theta_0; a_2, s_2)}{\pi(\theta_0; a_2, s_2)^2} \cdot g_1\right]\right]$$

$$= \mathbb{E}\left[(k(s_1', s_2') + k(s_2', s_1'))\left[-\frac{\partial_{\theta_0} \pi(\theta_0; a_1, s_1)}{\pi(\theta_0; a_1, s_1)^2}g_1g_2 - \frac{\partial_\theta \pi(\theta_0; a_2, s_2)}{\pi(\theta_0; a_2, s_2)^2}g_1g_2\right]\right]$$

$$- \mathbb{E}\left[(k(s_1', s_2') + k(s_2', s_1'))\left[\frac{\partial_\eta \omega(\eta_0; s_1')}{\pi(\theta_0; a_1, s_1)}g_2 + \frac{\omega(\eta_0; s_2')}{\pi(\theta_0; a_2, s_2)}g_1\right]\right]$$

$$\triangleq \mathbb{E}\left[-\partial_\eta G(\eta_0, \theta_0; (x_1, x_2))\partial_\theta S(\theta_0; (x_1, x_2))\right] + \mathbb{E}\left[(k(s_1', s_2') + k(s_2', s_1'))(H_1 + H_2)\right].$$

Here, we denote $H_1 = \frac{\partial_\eta \omega(\eta_0; s_1')}{\pi(\theta_0; a_1, s_1)} g_2$, $H_2 = \frac{\omega(\eta_0; s_2')}{\pi(\theta_0; a_2, s_2)} g_1$. Note that

$$\mathbb{E}[g_2 | a_1, s_1, s_1', s_2'] = \mathbb{E}\left[ \left( \omega(\eta_0; s_2) \frac{\pi(a_2 | s_2)}{\pi(\theta_0, a_2, s_2)} - \omega(\eta_0, s_2') \right) | a_1, s_1, s_1', s_2' \right] = 0,$$

$$E[g_1 | a_2, s_2, s_1', s_2'] = \mathbb{E}\left[ \left( \partial_\eta \omega(\eta_0; s_1) \frac{\pi(a_1 | s_1)}{\pi(\theta_0; a_1, s_1)} - \partial_\eta \omega(\eta_0; s_1') \right) | a_2, s_2, s_1', s_2' \right] = 0.$$

Therefore, $\mathbb{E}[(k(s_1', s_2') + k(s_2', s_1')) (H_1 + H_2)] = 0$. So we obtain (8). $\qquad \square$

### B.3  PROOF OF COROLLARY 1

**Corollary 1.** *Let $N$ be the number of $(s, a, s')$ tuples in the data. Under Assumptions 1 and 2,*

$$\mathbb{E}[(\hat{\eta} - \eta_0)^2] = O\left( \frac{1}{N} \right)$$

.

*Proof.* In the prove of Theorem 1, we see that

$$\frac{1}{N^2} \sum_{1 \le i, j \le N} \partial_\eta G(\eta_0, \theta_0; (x_i, x_j)) + K_1(\hat{\eta} - \eta_0) + K_2(\hat{\theta} - \theta) = o_p(1).$$

with $K_1 = \mathbb{E}\left[ \partial_\eta^2 G(\eta_0, \theta_0; (x_1, x_2)) \right]$ and $K_2 = \mathbb{E}\left[ \partial_\theta \partial_\eta G(\eta_0, \theta_0; (x_1, x_2)) \right]$. Therefore,

$$\mathbb{E}[(\hat{\eta} - \eta_0)^2] \le 2K_1^{-2} \left( K_2^2 \mathbb{E}[(\hat{\theta} - \theta_0)^2] + \mathbb{E}\left[ \left( \frac{1}{N^2} \sum_{1 \le i, j \le N} \partial_\eta G(\eta_0, \theta_0; (x_i, x_j)) \right)^2 \right] \right).$$

We assume that CLT holds for the maximum likelihood estimator $\hat{\theta}$, i.e. $\mathbb{E}[(\hat{\theta} - \theta_0)^2] = O(1/N)$. Besides, as $\mathbb{E}[\partial_\eta G(\eta_0, \theta_0; (x_i, x_j))] = 0$, under Condition 4 of Assumption 2, , we can apply the central limit theorem (for stationary Markov chain) and have

$$\mathbb{E}\left[ \left( \frac{1}{N\sqrt{N}} \sum_{1 \le i, j \le N} \partial_\eta G(\eta_0, \theta_0; (x_i, x_j)) \right)^2 \right] = O(1).$$

$\qquad \square$

## C  PROOFS OF THEORETIC RESULTS FOR EMP

**Propostion 1.** *The state-action-next-station tuples generated by $\pi_M$ follows the distribution $d_M(s, a, s')$. As a consequence, $d_M(s)$ is the stationary state distribution generated by $\pi_M$.*

*Proof.* It suffices to check that for any $s'$,

$$d_M(s') = \sum_{s, a} d_M(s) \pi_M(a | s) T(s' | a, s). \qquad (9)$$

For each behavior policy $\pi_j$, we have

$$d_{\pi_j}(s') = \sum_{s, a} d_{\pi_j}(s) \pi_j(a | s) T(s' | a, s).$$

Therefore,

$$\sum_j w_j d_{\pi_j}(s') = \sum_{s, a} \sum_j w_j d_{\pi_j}(s) \pi_j(a | s) T(s' | a, s).$$

Note that the left hand side is simple $d_M(s')$. In the right hand side,

$$\sum_j w_j d_{\pi_j}(s) \pi_j(a | s) = \left( \sum_k w_k d_{\pi_k}(s) \right) \frac{\sum_j w_j d_{\pi_j}(s)}{\sum_k w_k d_{\pi_j}(s)} \pi_j(a | s) = d_M(s) \pi_M(a | s),$$

and then we immediately obtain (9). $\qquad \square$

**Propostion 2.** *The average award satisfies*

$$R_\pi = E_{(s,a)\sim d_M} \left[ \omega_M(s) \frac{\pi(a|s)}{\pi_M(a|s)} r(s,a) \right].$$

*Proof.*

$$\mathbb{E}_{(s,a)\sim d_M} \left[ \omega(s) \frac{\pi(a|s)}{\pi_M(a|s)} r(s,a) \right] = \sum_{s,a} \omega(s) \frac{\pi(a|s)}{\pi_M(a|s)} r(s,a) \sum_j w_j d_{\pi_j}(s) \pi_j(a|s)$$

$$= \sum_{s,a} \omega(s) \frac{\pi(a|s)}{\pi_M(a|s)} r(s,a) d_M(s) \pi_M(s) = \sum_{s,a} d_M(s) \omega(s) \pi(a|s) r(s,a) = R_\pi.$$

$\square$

**Theorem 2.** *Under the same conditions of Theorem 1, if $\omega(\tilde{\eta}; s)$ and $\omega(\hat{\eta}; s)$ are the stationary distribution correction learned from (6) and from the same min-max problem but with exact value of $\pi_M$, then, asymptotically*

$$E[(\hat{\eta} - \eta_0)^2] \leq E[(\tilde{\eta} - \eta_0)^2].$$

*As a result, $E[(\hat{\eta} - \eta_0)^2] = O\left(\frac{1}{N}\right)$.*

*Proof.* The proof follows immediately from that of Theorem 1. In particular, assume $\pi_M \in \{\pi(\theta; a, s) : \theta \in \mathcal{E}_\theta\}$ and the estimated $\hat{\pi}_M = \pi(\hat{\theta}; \cdot)$ is obtained via

$$\hat{\theta} = \arg\max_\theta \sum_j \sum_{n=1}^{N_j} \log(\pi(\theta; s_{j,n}, a_{j,n})).$$

The rest part of the proof follows the same argument in the proof of Theorem 1. $\square$

## D EXPERIMENTAL DETAILS

### D.1 ENVIRONMENT DESCRIPTION

**Taxi** (Dietterich & G, 2000) is a $5 \times 5$ grid world simulating a taxi movement. Six actions are contained in Taxi: moves North, East, South, West, pick up and drop off a passenger. A reward of 20 is received when it picks up a passenger or drops her/he off at the right place, and a reward of -1 for each time step. The passengers are allow to randomly appear and disappear at every corner of the map at each time step. The $5 \times 5$ grid size yields 2000 states in total ($25 \times 24 \times 5$, corresponding to 25 taxi locations, 24 passenger appearance status and 5 taxi status (empty or with one of 4 destinations)).

**Gridworld** (Thomas & Brunskill, 2016) is a $4 \times 4$ grid world which including one reward state, one terminate state and one fire state and thirteen normal state. Four action can be taken in this environment: up, down, left and right. A reward of -1 will be received while the agent in normal states, 1 reward is obtained in reward state, 100 reward is got in terminate state and -11 reward will got in fire state.

**SinglePath** has 5 states, 2 actions. The agent begins in state 0 and both actions either take the agent from state n to state n + 1 or cause the agent to remain in state n. If the agent arrives at a new state, it will receive a +1 reward, otherwise it will get a 1 reward.

**Pendulum** has a continuous state space of $\mathcal{R}^3$ which describes the triangle of and a action space of $[-2, 2]$.

### D.2 MIXTURE POLICY ESTIMATION

For the discrete environments Taxi, Gridworld and SinglePath, the MLE estimate (5) coincide with count-frequency, and therefore, we directly estimate

$$\hat{\pi}_M(a|s) = \frac{\sum_j \sum_{n=1}^{N_j} 1(a_{j,n} = a, s_{j,n} = s)}{\sum_j \sum_{n=1}^{N_j} 1(s_{j,n} = s)}.$$

For the continuous environment Pendulum, we use a neural network to model the policy. In detail, we train a two-layer MLP neural network to estimate the policy. The size of the two hidden layers are both 32 with the learning rate 0.001 and tanh activation function. We use MEL (5) and Adam optimizer to train the neural network with batch size 128.

### D.3    KL-EMP

In EMP algorithm, the proportion of samples from policy $\pi_j$ in the data buffer, $w_j$, is fixed. In an ad-hoc way, we optimize the weights $w_j^{KL}$ according to the KL-divergence between the behavior policy $\pi_j$ and the target policy. Then, we will generate a new data buffer as follows. First, we sample $j \in \{1, 2, ..., m\}$ with probability $w_j^{KL}$. Then, given $j$, we sample uniformly from the subgroup of data generated by $j$. Note that, to implement KL-EMP, one do not need to know the exact value of $\pi_j$, but one need to know which data are generated from which policy.

In the numerical experiment, we use the following formula to compute $w^{KL}$ for finite state space:

$$
\begin{aligned}
w_j^{KL} &= \frac{\sum_{s \in S} \mathbf{1}(j = \arg\min_{1 \leq k \leq m} D_{KL}(\pi(\cdot|s)||\pi_k(\cdot|s)))}{\sum_{i=1}^{m} \sum_{s \in S} \mathbf{1}(i = \arg\min_{1 \leq k \leq m} D_{KL}(\pi(\cdot|s)||\pi_k(\cdot|s)))} \\
&= \frac{\sum_{s \in S} \mathbf{1}(j = \arg\min_{1 \leq k \leq m} D_{KL}(\pi(\cdot|s)||\pi_k(\cdot|s)))}{|S|}
\end{aligned}
\tag{10}
$$

To implement this method for infinite or continuous state space, we replace the set of all possible states $S$ in (10) with the set of all states that has been visited in the data buffer. Besides, the behavior policy $\pi_k$ is unknown, to estimate the KL-divergence $D_{KL}(\pi(\cdot|s)|\pi_k(\cdot|s))$, we use a neural network to learn $\pi_k$ from the subgroup of data that are generated from policy $\pi_k$.

The numerical results show that using the KL weights $\{w_j^{KL}\}$ could achieve smaller MSE compared to using $\{w_j\}$ as given by the data sample. We believe this approach deserves more careful analysis in future research studies.

### D.4    ADDITIONAL EXPERIMENT RESULTS

Note that EMP, EMP (single) and KL-EMP all have their policy-aware analogues. In order to test the variance reduction effect of policy estimation, we implement the policy-aware version for each EMP-type algorithm and compare their performances.

The policy-aware version of EMP (single) is naive BCH, in which we first apply BCH to each behavior policy and then return the estimation average.

The policy-aware version of EMP is named as BCH (pooled). In BCH (pooled), the corresponding min-max problem formation is

$$
\min_{\omega} \max_{f} \mathbb{E}_{(j,s,a,s') \sim \mathcal{D}} \left[ \left( \omega(s') - \omega(s) \frac{\pi(a|s)}{\pi_j(a|s)} \right) f(s') \right]^2 .
\tag{11}
$$

They both pool the data from different behavior policies together and the main difference is that BCH (pooled) uses the exact behavior policies.

The policy-aware version of KL-EMP is called BCH (KL-polled). The main difference between BCH (KL-polled) and BCH (polled) is that BCH (KL-polled) utilizes KL-divergence to optimize the weights.

Comparison results are shown in Figure 5. We observe that the partially policy-agnostic methods consistently outperform their policy-aware analogues.

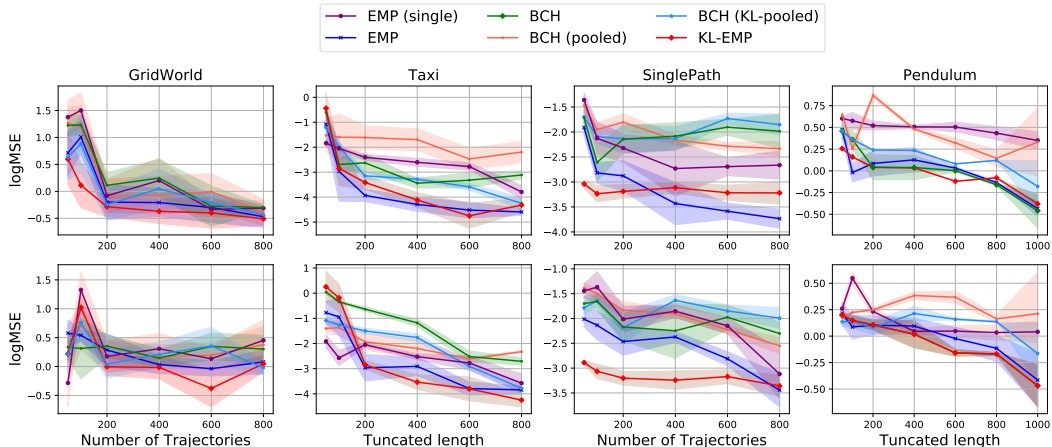

Figure 5: Results of policy-aware OPPE methods (BCH, BCH (pooled) and BCH (KL-pooled)) and their corresponding partially policy-agnostic version (EMP (single), EMP and KL-EMP ) across continuous and discrete environments with average reward.

