# OpenReview forum: "Infinite-horizon Off-Policy Policy Evaluation with Multiple Behavior Policies"
_ICLR.cc/2020/Conference — Accept (Poster)_

### Official Review · AnonReviewer1 · 2019-10-22
**Official Blind Review #1**

**Rating:** 6

**Review:**

  *Synopsis*:
  The main contribution of this paper is the development of estimated mixture policy (EMP), which takes ideas from the new off-policy policy evaluation infinite horizon estimators (i.e. Liu) and from a recent development in more traditional importance weight approaches using regression importance sampling (i.e. Hannah). This new method provides a nice extension of Liu's algorithm to many policies, and to when the policy is unknown. The paper provides some nice analysis inspired by Hannah. Finally, they provide empirical results.


  *Review*:

  While I think the method has potential interest to the community, I found the empirical results lacking (particularly in missing competitors). I also have some concerns about the theory as there seems to be many typos and consistency issues making much of it hard to follow. Overall, I think this paper is not quite up for publication, but if the authors address my consistency/missing proofs issues and provide a comparison to DualDice I will increase my score.

  1. I don't find the reason provided for not including DualDICE compelling and think it is an important competitor here, as there are many similarities between the two methods.
     - It would also be interesting to reproduce the results provided by Liu et al with the model based approach, and the on-policy oracle. I don't think these are as pressing as DualDice but still interesting.

  2. There are many consistency issues with respect to notation, and some odd notation choices as compared to the rest of the literature:
     - What is script \epsilon in the equation in 2.1? It looks like it should be an expectation over d_\pi, but I've never seen this notation before.
     - The indicies of sums and sets often change between one based and zero based indexing. This should be unified (preferably to one based). For example, section 2.1 \pi_j(j=0,1,2,...,m) and m=1 for one policy doesn't work. Either \pi_j(j=0,1,2,...m-1) or m=0. This occurs throughout the proofs in the appendix as well.
     - What is script \Epsilon_\theta? Do you mean script F_\theta? And then what does it mean for \theta_0 \in E_1? There seems to be many definitions missing.

  2.5. There are also some issues with the presentation of the theory over the consistency issues already mentioned.
     - The assumptions and conditions for the theorems presented in the main text should be clearly specified in the theory statement.
     - The proof to theorem 2 (i.e. in the appendix) should be provided.
     - The proof of proposition 4 seems to be missing as well.

  *Questions*
  Q1 What are the properties of SADL? Why was this used instead of DualDICE in the comparison?

  Q2 How would BCH do if we used the mixture policy as the behavior policy in the multiple behavior policy case? How would it compare to your method? This could be an interesting comparison, just to test if the lower MSE argument holds up in the multiple behavior policy case.

  Q3 What is the meaning of partially-policy-agnostic? It is unclear to me how it is different from the policy-agnostic approaches. If all that is different is you are estimating the behavior to use in the usual infinite-horizon approach, should this be in a separate category from policy-agnostic approaches? (I would say probably not, but I think you could make a case for it).

  Q4 "Then, a single (s,a,s') tuple simply follows the marginal distribution...". Is this trivially true?

  *Minor comments not taken into account in the review*
  - section 2.1 "target policy \pi via a pre-collected..." -> remove "a"
  - The layout of the related works section is a bit hard to follow.
  - "Recently, Nachum et. al. (2019) proposes DualDice": proposes->proposed
  - "by their estimated values in two folds": do you mean in two ways? This is unclear.
  - Section 3.1: "notation abusion" -> notation abuse
  - Equation right after equation 1: "d_\pi_0(s)\pi(a|s)" -> "d_\pi_0(s)\pi_0(a|s)"
  - "The derivation of kernel method are put in..." -> "The derivation of the kernel method is put in..."
  - "we need introduce more notation" -> "we need to introduce more notation"
  - Section 4: "detailed description on the data sample": "on"->"of"
  - "In this light by pooling the data together...." These two sentences should be put together.
  - I would like if your theorems were restated in the appendix, for ease of reading.

-----------
Post Rebuttal

I'd like to thank the author for their thorough response! Given the major additions to the paper including empirical comparisons and clarity for the theory I've decided to update my score to reflect my new feelings (i.e. to a 6). I think this paper is well worth accepting in its current form.


**Experience Assessment:**

I have read many papers in this area.

**Review Assessment: Checking Correctness Of Derivations And Theory:**

I carefully checked the derivations and theory.

**Review Assessment: Checking Correctness Of Experiments:**

I carefully checked the experiments.

**Review Assessment: Thoroughness In Paper Reading:**

I read the paper thoroughly.

---

> ### Author Response · Authors · 2019-11-15
> **Response to Review #1**
>
> Response to Major Comments:
> 1.
> - We got the released code and carried out a comparison study with DualDICE.
> - We implemented  a variation of BCH method by Liu et al for mutliple-behavior-policy cases, called BCH (pooled), using the exact value of all behavior policies. We now report the comparison results of our method EMP, BCH and DualDICE in both single- and multi-behavior-policy settings.
> - We use the estimation results by on-policy oracle with large trajectory length and number as the “true-value”. This is how we compute the MSE for EMP and other benchmark OPPE methods.
>
> 2.
> - We have correct this typo on page 2.
> - We have unified the notation so that all indicies start from 1.
> - We have correct these typos. $\mathcal{E}_\theta$ is the correct notation for parameter space.
> - We also went through the mathematical part of the paper and corrected the typos we have found.
>
> 2.5
> - We now cite the assumptions in the statement of the theorems. The detailed assumptions are stated and explained in Appendix B. We didn’t directly include the full assumptions in the statement of the theorem mainly because of the space limit. Most of the assumptions actually involve technical details about the kernel method that solves the min-max problems, and they need space of half to one page.
> - We have removed the SADL algorithm in the revision, as it was used as an substitute of DualDICE in the previous version of the paper. So we also removed (the original) Theorem 2, which follows a very similar proof as the kernel-method derivation in Liu et. al.
> - Proposition 4 (now 3) has been proved in Veach & Guibas (1995). We include it mainly for self-containedness and refer to the original paper for the proof. We now cite Veach & Guibas (1995) more explicitly in the statement of Proposition 3.
>
> Response to Questions:
> Q1 What are the properties of SADL? Why was this used instead of DualDICE in the comparison?
> - We are not aware of the DualDICE code releasing when we first submitted this paper. So we use SADL as a substitute of DualDICE. They are both policy-agnoistic and learn the state-action joint distribution correction.
>
> Q2 How would BCH do if we used the mixture policy as the behavior policy in the multiple behavior policy case? How would it compare to your method? This could be an interesting comparison, just to test if the lower MSE argument holds up in the multiple behavior policy case.
> - We implemented a multiple-behavior-policy version of Liu et al, using the information of all behavior policies. We call it BCH (pooled) and report the comparison results of our method EMP, BCH (pooled) and DualDICE.
>
> Q3 What is the meaning of partially-policy-agnostic? It is unclear to me how it is different from the policy-agnostic approaches. If all that is different is you are estimating the behavior to use in the usual infinite-horizon approach, should this be in a separate category from policy-agnostic approaches? (I would say probably not, but I think you could make a case for it).
> -We call EMP a partially policy-agnostic method in the sense that, although EMP does not require any information on the “physical” behavior policies, it learns a “virtual” policy, which is the mixture policy, defined formally in Section 4.1, and contains aggregated information about the “physical” behavior polices. As a consequence, the accuracy of the “virtual” policy learning (conceptually, whether the algorithm can effectively extract the aggregated information about the behavior policies) will affect the performance of EMP. We now add this explanation to the introduction part.
>
> Q4 "Then, a single (s,a,s') tuple simply follows the marginal distribution...". Is this trivially true?
> - We added a new Subsection 4.1 to more formally state our assumptions on the sample data collected from different behavior policies and the relevant distributions.
>
> Response to Minor Comments:
> - We have corrected all the typos accordingly. As to the related work part, we added a few sentences of explanation at several places to improve the logic flow.

---

### Official Review · AnonReviewer2 · 2019-10-23
**Official Blind Review #2**

**Rating:** 6

**Review:**

After rebuttal:
Thank author for the clarification. The new version looks better and I tend to accept the paper in the current version.
=========
This paper provides a algorithm to solve infinite horizon off policy evaluation with multiple behavior policies by estimate a mixed policy under regression, and follows the same method of BCH. The intuition of using an estimated policy comes from Hanna et al. (2019) which shows that an estimated policy ratio can reduce variance even it introduces additional bias. The authors provide theoretical proof on that and arguing that their method is not worse than BCH one. Empirical results show that in general their method performs as good as previous baseline. I believe this method is novel and natural and worth investigating.

Technical Concerns:
The major concern I have is in continuous case, it is almost impossible to pre-assume a model for learning the mixed policy $\hat{\pi_0}$. For example, if the sample policies $\pi_j$ are all Gaussians, then $\pi_0$ according to equation (4) would be a complicated mixture distribution (not even a Gaussian mixture since it involves ratio of $d_{\pi_j}$ which is hard to compute). If the model is not precise, we cannot achieve the bias/variance tradeoff where the bias introduce by model mismatching can be arbitrarily large.
And according to the experimental details in appendix E, I didn't find any useful model assumption to address that issue. So my question would be: what model are you using when doing regression for $\pi_0$?

Clarity Concern:
The written of the paper is not satisfactory. The major contribution should be highlight in section 4, which from my first time reading is very unclear. The key observation of equation (4) uses a recursive definition, where we define $\pi_0$ using an undefined $d_{\pi_0}$. I think you should rewrite $d_{\pi_0}$ as the weighted summation of $d_{\pi_j}$. And you should avoid repeating similar equation like (2) (3) (6) and (7) where you can cite equation (2) in general or use a short notation for that equation, otherwise it is hard to contract the contribution of the paper.
The experiment part is also unclear. Here's some questions: 1) How many repetitions you apply for each figure? It seems not smooth enough. 2) Which estimator you use for you regression? Maximum Likelihood Estimation? Which model you are using for each environment (I know for tabular MLE is count based)? 3) What is $\pi_k$ in section E.3 equation (12)? How do you compute KL divergence in this equation for empirical distribution?

Some minor issues:
1. You should replace 'for all' to $\forall$ when writing equation, like equation before (2), equation (6) and equation in proposition 4.
2. You'd better to separate legend with figure in order to make the legend larger and figure clearer.

Overall, I think this work is very novel and natural, but should give more consideration on the model selection. I tend to reject the paper by the current version and encourage the authors to submit to another conference after the revision.

**Experience Assessment:**

I have published one or two papers in this area.

**Review Assessment: Checking Correctness Of Derivations And Theory:**

I carefully checked the derivations and theory.

**Review Assessment: Checking Correctness Of Experiments:**

I carefully checked the experiments.

**Review Assessment: Thoroughness In Paper Reading:**

I read the paper thoroughly.

---

> ### Author Response · Authors · 2019-11-15
> **Response to Review #2**
>
> Response to Technical Concerns:
> In our experiment, we use a neural network to approximate the mixture policy in continuous case and estimate the model by MLE. We have added a paragraph explaining this for the general algorithm in Section 4.2 and for the numerical experiment in Appendix D.2. We agree with the reviewer that when model is not precise, the bias will overwhelmed the variance reduction. We explicitly explained that the performance of EMP relies on the accuracy of policy learning in the introduction part. We think the problem of model uncertainty is interesting and more challenging, and probably requires a different theoretic framework, such as robust optimization, so we will leave this for further study. Therefore, in most part of the paper, to study the effect of policy learning on OPPE performance, we would like to focus on the cases that the policy can be well approximated by some parametric model, especially for theoretic analysis.
>
> Response to Clarity Concerns:
> - We have modified Section 4 according to the comments of the reviewers. We added a subsection to more formally define the mixture policy pi_M and the mixture distribution d_M (to better distinguish from the single-behavior policy, we have also changed the notation.) We also provided additional theoretic properties of \pi_M and d_M to provide more intuition behind the algorithm design.
> - We have removed equation (3) and (6). But we keep equation (7) (now becomes (6)) to make the description of EMP algorithm more self-contained.
> - We have also modified the experiment part. First, we added new comparison results to the state-of-art policy-agnostic method DualDICE. Second, we reorganized experiment part to make it more consistent with the theoretic analysis, hope this could convey clearer messages of the numerical experiments.
>
> Response to Questions:
> 1)How many repetitions you apply for each figure? It seems not smooth enough.
> -we have increased the number of repetitions by 3 times and updated the numerical results.
> 2)Which estimator you use for you regression? Maximum Likelihood Estimation? Which model you are using for each environment (I know for tabular MLE is count based)?
> - For the three discrete environment, we used count-frequency to estimate the policy. For the continuous environment, we used a neural network to model the policy and MLE to estimate the model parameters. We have added a paragraph explaining our model and estimation methods in Appendix D.2. (E.2 in the previous version).
> 3)What is in section E.3 equation (12)? How do you compute KL divergence in this equation for empirical distribution?
> - Equation (12) is used to defined the adjusted proportion of data from each policy. It involves the KL-divergence, which is computed by estimating the behavior policy by a parametric model. We have added some explanation in Section D.3 (E.3 in the previous version.)
>
> Response to Minor Issues:
> 1.You should replace 'for all' to when writing equation, like equation before (2), equation (6) and equation in proposition 4.
> - We have replaced ‘for all’ with ‘\forall’ in the equations.
> 2.You'd better to separate legend with figure in order to make the legend larger and figure clearer.
> - We have changed the format of figures accordingly.

---

### Official Review · AnonReviewer3 · 2019-10-25
**Official Blind Review #3**

**Rating:** 3

**Review:**

The authors propose here a method for off-policy policy evaluation (OPPEval), to use the reinforcement reinforcement learning on infinite horizon problems from previously-collected trajectories.

The authors frame their work that much of the focus in the OPPEval field has been on, as they call, "policy-aware" methods that use information from the policy  to improve estimates  to cope with the mismatch between then behaviour and the estimated target policy (such as IS, WIS, etc) when computing state stationary distribution. These contrast "policy agnostic" methods (DualDICE, Nachum et al, 2019) that suffer from the curse of dimensionality in estimating the much higher dimensional state-action stationary distributions but do not depend on policy information.
The manuscripts novelty rests in a comparing and relating  these agnostic/aware approaches and propose a partially policy-agnostic method (EMP) that strives to combine advantages from both approaches by following a mixture approach (effectively a mixture of weighted policies). The authors provide a derivation of their methods bounds and show that their method outperforms policy-aware methods as well as policy-agnostic methods. In the comprehensive experiments they compare recent methods by Liu et al (BCH) and WIS (I suppose they mean weighted importance sampling following Prenup et al 2000, as no citation given) ), as well a a new policy-agnostic method they propose here (SADL). In all cases the results  favour the proposed new method (EMP). The results advance the field by providing a pathway to improved estimation results (lower uncertainty) by using policy mixtures.
While I have not checked the derivations line-by-line the results are consistent and interesting, although not entirely clear to me why this is an important contribution to a representational learning conference.

A key question to this paper (and the OPPEval field) is to evaluate their methods  more consistently in closed-loop agent experiments - after training on the historical data. Perhaps for a representational learning conference this would be more appropriate way to convince one of the strength of the results.


**Experience Assessment:**

I have published in this field for several years.

**Review Assessment: Checking Correctness Of Derivations And Theory:**

I did not assess the derivations or theory.

**Review Assessment: Checking Correctness Of Experiments:**

I carefully checked the experiments.

**Review Assessment: Thoroughness In Paper Reading:**

I read the paper at least twice and used my best judgement in assessing the paper.

---

> ### Author Response · Authors · 2019-11-15
> **Response to Review #3**
>
> - We have added the missing citation Prenup et al 2000.
>
> -  To certain extend, our method shows the performance improvement of OPPEval algorithm by using certain representation model (especially in continuous case) to learn the specific information about the underlying behavior policies from data. Besides, there are applications where policy evaluation is the ultimate goal in itself [1]. Policy evaluation algorithms are also important to study because they are often key parts of larger algorithms where the ultimate goal is to find an optimal policy (one such example is the class of actor-critic algorithms, see [2] for a survey). We feel that ICLR seems to be expanding its scope and covering more general topics in ML and the topic of this paper fits this trend.
>
> [1] P. Balakrishna, R. Ganesan, and L. Sherry, “Accuracy of reinforcement learning algorithms for predictingaircraft taxi-out times: A case-study of tampa bay departures,”Transportation Research Part C: EmergingTechnologies, vol. 18, no. 6, pp. 950–962, 2010.
>
> [2]I. Grondman, L. Busoniu, G. A. Lopes, and R. Babuska, “A survey of actor-critic reinforcement learning:Standard and natural policy gradients,”IEEE Transactions on Systems, Man, and Cybernetics, Part C(Applications and Reviews), vol. 42, no. 6, pp. 1291–1307, 2012

---

### Author Response · Authors · 2019-11-15
**Revision Summary**

1. We carry out a comparison study with DualDice in the revision.
2. We re-organize the theoretic derivation to highlight the key ideas of EMP algorithm. In particular, we explain more explicitly the definition of the mixture policy in the multiple-behavior-policy setting, and its connection with the sample data distribution, to better explain our notion of "partially policy-agnostic".
3. We re-organize the experiment part to make it more consistent with the theoretic analysis results. We also did more experiment repetitions to get better MSE estimation.

---

### Decision · Program_Chairs · 2019-12-19

**Decision:**

Accept (Poster)

**Comment:**

The authors present a method to address off-policy policy evaluation in the infinite horizon case, when the available data comes from multiple unknown behavior policies.  Their solution -- the estimated mixture policy -- combines recent ideas from both infinite horizon OPE and regression importance sampling, a recent importance sampling based method.  At first, the reviewers were concerned about writing clarity, feasibility in the continuous case, and comparisons to contemporary methods like DualDICE.  After the rebuttal period, the reviewers agreed that all the major issues had been addressed through clarifications, rewriting, code release, and additional empirical comparisons.  Thus, I recommend to accept this paper.